# A Novel Approach to 3D-DOA Estimation of Stationary EM Signals Using Convolutional Neural Networks

**DOI:** 10.3390/s20102761

**Published:** 2020-05-12

**Authors:** Dong Chen, Young Hoon Joo

**Affiliations:** School of IT Information and Control Engineering, Kunsan National University, Gunsan 54150, Korea; chendongddd@sina.com

**Keywords:** direction-of-arrival (DOA), convolutional neural network (CNN), Gaussian noise, non-Gaussian noise, uniform triangular array (UTA)

## Abstract

This paper proposes a novel three-dimensional direction-of-arrival (3D-DOA) estimation method for electromagnetic (EM) signals using convolutional neural networks (CNN) in a Gaussian or non-Gaussian noise environment. First of all, in the presence of Gaussian noise, four output covariance matrices of the uniform triangular array (UTA) are normalized and then fed into four neural networks for 1D-DOA estimation with identical parameters in parallel; then four 1D-DOA estimations of the UTA can be obtained, and finally, the 3D-DOA estimation could be obtained through post-processing. Secondly, in the presence of non-Gaussian noise, the array output covariance matrices are normalized by the infinity-norm and then processed in Gaussian noise environment; the infinity-norm normalization could effectively suppress impulsive outliers and then provide appropriate input features for the neural network. In addition, the outputs of the neural network are controlled by a signal monitoring network to avoid misjudgments. Comprehensive simulations demonstrate that in Gaussian or non-Gaussian noise environment, the proposed method is superior and effective in computation speed and accuracy in 1D-DOA and 3D-DOA estimations, and the signal monitoring network could also effectively control the neural network outputs. Consequently, we can conclude that CNN has better generalization ability in DOA estimation.

## 1. Introduction

Direction-of-arrival (DOA) estimation refers to a signal processing technique for processing the incoming wave signals received by arrays and estimating the position of the signal emitters. It has been widely applied in many fields in recent years, such as radar, sonar, electronic monitoring, mobile communication, and seismic research [1,2,3]. In the presence of Gaussian noise, conventional 1D-DOA estimation methods of stationary electromagnetic (EM) signals include MUSIC [4,5], MVDR [6,7], TLS-ESPRIT [8,9,10], etc. MUSIC and MVDR require the spectral peak search and achieve high accuracy. However, they suffer from high computational complexity. TLS-ESPRIT avoids the spectral peak search and requires less computation. Also, the estimation accuracy is not high enough. As for 3D-DOA estimation in Gaussian noise environment, the 3D-MUSIC algorithm is still the most popular one at present. It needs to perform a cyclic search for the three parameters of the azimuth, elevation, and radial distance. Therefore, its computational complexity is extremely high. In the presence of non-Gaussian noise, conventional 1D-DOA estimation methods of stationary EM signals include FLOM [11], PFOM [12], CRCO [13], COBU [14], etc. Besides the spectral peak search, they also need a complex preprocessing stage and require high computational complexity. So far, to the best of our knowledge, there is no effective method regarding 3D-DOA estimation in non-Gaussian noise environment. Therefore, the high computational cost is the main problem for these conventional algorithms in real-time applications.

In order to overcome the computational complexity of conventional methods, neural networks have been developed to DOA estimation. References [15,16,17,18] using the support vector machine (SVM) [19] and references [20,21,22,23] using the multi-layer perceptron (MLP) [24] formulate DOA estimation as a classification problem, and they lead to the discrete outputs of neural networks. References [25,26,27,28] use a radial basis function (RBF) [29] to formulate DOA estimation as a regression problem, and they lead to the continuous outputs of neural networks. These DOA estimation methods based on neural networks effectively overcome the computational complexity of conventional methods and have high estimation accuracy. However, most of the existing DOA estimation methods based on neural networks have only focused on 1D or 2D estimation in the presence of Gaussian noise, and their ranges of practical applications are limited. In addition, in the presence of non-Gaussian noise, these neural networks fail to obtain effective input features and perform correct DOA estimation due to the non-convergence of the second-order moments of array outputs. Therefore, the dimension must be extended to 3D, and an effective solution must also be proposed in non-Gaussian noise. In the last few years, with the rapid development of artificial intelligence, the convolutional neural network (CNN) has been increasingly studied by researchers. CNN is a well-known deep learning architecture inspired by the natural visual perception mechanism of living creatures. The applications of CNN include computer vision, speech, natural language processing, etc. For details about CNN, see Gu et al. [30] and the references therein. 

Motivated by the aforementioned analysis, in this paper, we present a novel 3D-DOA estimation method for stationary EM signals using CNN in a Gaussian or non-Gaussian noise environment. To do this, we propose a 1D-DOA estimation neural network based on CNN and extend the dimension from 1D to 3D to achieve 3D-DOA estimation utilizing the proposed uniform triangular array (UTA). In the presence of non-Gaussian noise, we utilize the infinity-norm normalization to propose a solution that provides appropriate input features for the neural network. In addition, a signal monitoring network is also proposed to control the outputs of the neural network to avoid misjudgments of the neural network. Then, we propose a novel 3D-DOA estimation method of stationary EM signals. Finally, some numerical examples prove the superiority and effectiveness of the proposed method.

The main contributions of this paper are as follows: (1) CNN is introduced into DOA estimation of EM signals; (2) the UTA for 3D-DOA estimation based on neural networks is proposed; (3) by means of the infinity-norm normalization preprocessing, the neural network can achieve DOA estimation in the presence of non-Gaussian noise; (4) the signal monitoring neural network of EM signals is proposed.

The remainder of this paper is organized as follows: In Section 2, we review Gaussian noise and non-Gaussian noise, defining the problem of interest; in Section 3, we present the architecture and design process of the 3D-DOA estimation model in the presence of Gaussian noise, proposing a solution using the infinity-norm normalization for the scenario of non-Gaussian noise; finally, Section 4 demonstrates simulation results, and Section 5 summarizes the conclusions and future work. The main notations used in this paper are listed in Table 1. Other terms used in this paper follows the general notations unless otherwise stated.

## 2. Preliminary and Problem Formulation

### 2.1. Noise Model

At present, most of the DOA estimation algorithms based on neural networks assume that the noise environment is Gaussian noise, which refers to a class of noises whose probability density function (PDF) obeys Gaussian distribution. Common Gaussian noises include fluctuation noise, cosmic noise, thermal noise, shot noise, etc. Figure 1 shows the time-domain waveform of Gaussian distribution with a mean of 0 and a variance of 1, and Figure 2 shows its PDF.

However, many signals and noises encountered in practice are decidedly non-Gaussian, for example, low-frequency atmospheric noise, underwater acoustic signals, and many types of human-made noises. An important class of impulse noise encountered in DOA estimation can be modelled by *α*-stable distribution [14], which is an extremely flexible modelling tool. It is pity that there exists no closed-form expression for the PDF of *α*-stable distribution except for Gaussian and Cauchy distributions. The *α*-stable distribution is generally defined by its characteristic function as:(1)$$\varphi (t)=\exp\{j\mu t-\gamma {\left|t\right|}^{\alpha }[1+j\beta \mathrm{sgn}(t)\omega (t,\alpha )]\},$$
where:(2)$$\omega (t,\alpha )=\left\{ \begin{array}{l} \tan(\pi \alpha /2)\;\alpha \neq 1\\ (2/\pi )\log\left|t\right|\;\alpha =1 \end{array}\right.$$
and:(3)$$\mathrm{sgn}(t)=\left\{ \begin{array}{l} t/\left|t\right|\;t\neq 0\\ 0\;t=0 \end{array}\right.\;.$$

This has four parameters: the characteristic exponent *α* (0 < *α* ≤ 2), scale parameter *γ* (*γ* > 0), symmetry parameter *β* (−1 ≤ *β* ≤ 1), and location parameter *μ* (–∞ < *μ* < +∞). *α* determines the thickness of the tail of the distribution. *γ* (*γ* = 1 in this paper) determines the degree of discretization of the data samples, and it is similar to the variance of Gaussian distribution. *β* (*β* = 0 in this paper) determines the sign and degree of asymmetry about *μ* (*μ* = 0 in this paper), which is similar to the mean of Gaussian distribution. For details about the *α*-stable distribution, see Tian et al. [14] and the references therein. As shown in Figure 3, the time-domain waveform of *α*-stable distribution with *α* = 1.0 has more impulsive outliers compared with that in Figure 1. The smaller *α*, the more impulsive outliers and the stronger impulses. When *α* takes different values, Figure 4 shows the corresponding PDF of *α*-stable distribution and has obvious tails compared with that in Figure 2. The smaller *α*, the thicker the tail.

### 2.2. Problem Formulation

In order to achieve 3D-DOA estimation, we propose the UTA. Assume that a far-field stationary EM signal impings on the UTA, as shown in Figure 5. Three uniform linear arrays (ULA) *AB*, *BC*, and *CA* jointly constitute the UTA. The origin *O* of Cartesian coordinate locates at the midpoint of the altitude passing through the point *A*. The coordinates of the point *A*, *B*, and *C* are set to (*A_x_*, *A_y_*, 0), (*B_x_*, *B_y_*, 0), and (*C_x_*, *C_y_*, 0) respectively, and the estimated EM signal *s* is set to (*x*, *y*, *z*). The angles of *s* and *CA*, *s* and *CB*, *s* and *BA*, and *s* and *BC* are set as *θ_CA_*, *θ_CB_*, *θ_BA_*, and *θ_BC_**,* respectively, and the values of the four angles are all in the range of [0°, 180°]. 

Assume that every ULA consists of *M* omnidirectional undifferentiated sensors, and the distance between two sensors is *d*. The number of UTA sensors is 3 (*M* – 1). The single snapshot *M* × 1 observation vector of every ULA can be expressed as:(4)$$x(n)={\left[x_{1}(n),\cdots ,x_{M}(n)\right]}^{T}.$$

The *M* × 1 noise vector can be expressed as:(5)$$e(n)={\left[e_{1}(n),\cdots ,e_{M}(n)\right]}^{T}.$$

The *p* × 1 signal vector can be expressed as:(6)$$s(n)={\left[s_{1}(n),\cdots ,s_{p}(n)\right]}^{T},$$
where *p* is the number of signals. In the presence of Gaussian noise, the signal-to-noise ratio (SNR) is defined as $\mathrm{SNR}=10\mathrm{lg}(\sigma_{s}^{2}/\sigma_{e}^{2})$, where $\sigma_{s}^{2}$ denotes the signal power, and $\sigma_{e}^{2}$ signifies the noise power. The *M* × 1 steering vector can be expressed as: (7)$$A={\left[1,e^{-j2\pi \frac{d}{\lambda }\cos\theta },\cdots ,e^{-j(M-1)2\pi \frac{d}{\lambda }\cos\theta }\right]}^{T},$$
where *λ* denotes the wavelength of the carrier. The single snapshot *M* × 1 observation vector can also be expressed as [14]:(8)x(n)=As(n)+e(n).

The *M* × *N* observation matrix of all snapshots can be expressed as:(9)x=As+e,
where ***e*** and ***s*** denote the *M* × *N* noise matrix and *p* × *N* signal matrix, respectively, and *N* is the number of snapshots. The angle *θ* of the EM signal *s* and the ULA can be estimated by processing ***x***. Based on the proposed UTA, we could estimate a total of four angles.

Generally, 3D-DOA estimation is to consider the azimuth, elevation, and radial distance [31,32]. If the neural network is expected to estimate the three parameters directly, a large amount of training data must be combined from the three parameters to meet the demand of neural networks for data volume, but such a large amount of data is difficult to obtain. We significantly reduce the scale of the training set by dimension reduction. The problem addressed in this paper is to utilize a neural network to process four observation matrices in parallel and to synchronously estimate *θ_CA_*, *θ_CB_*, *θ_BA_*, and *θ_BC_*, and then to estimate the 3D Cartesian coordinate (*x*, *y*, *z*) of the stationary EM signal *s* through a simple post-processing, i.e., 3D estimation.

## 3. Proposed 3D-DOA Estimation Method in Gaussian or Non-Gaussian Noise Environment

In this section, first of all, we propose a 3D-DOA estimation model in the presence of Gaussian noise. The proposed 3D-DOA estimation model consists of three modules, as shown in Figure 6. The preprocessing module processes the UTA output data to obtain four normalized ULA output covariance matrices N-*R_CA_*, N-*R_CB_*, N-*R_BC_*, and N-*R_BA_*. The neural network module consists of a signal monitoring network and a DOA estimation network. The DOA estimation network consists of four parallel 1D-DOA networks with identical parameters. The inputs of the DOA estimation network are four normalized covariance matrices respectively, and its outputs correspond to *θ_CA_*, *θ_CB_*, *θ_BC_*, and *θ_BA_*. Despite low SNR or no incoming signals, still, the existence of noises will cause the DOA estimation network to generate outputs.

To avoid the noticeable misjudged outputs, the outputs of the model should be controlled by the monitoring network. The input of the monitoring network can be any normalized covariance matrix from the UTA, and the output value is 1 or 0. 0 indicates invalid incoming wave signals and no model outputs. 1 shows effective incoming wave signals and normal model outputs. The inputs of the post-processing module are the four output angles of the DOA estimation network, and its output is the EM signal position (*x*, *y*, *z*). In addition, if the noise environment is non-Gaussian, the array output data must be additionally preprocessed, as presented in Section 3.5. 

### 3.1. Preprocessing

The purpose of data preprocessing is to feed appropriate input features into the neural network. The array model has been presented in Section 2.2, and the sample covariance matrix of any ULA output can be expressed as [14]:(10)$$R_{xx}=E\{x(n)x^{H}(n)\}=APA^{H}+\sigma_{e}^{2}I,$$
where ***P*** = *E*{***s***(*n*)***s***^H^(*n*)}. The ideal covariance matrix is not available, but more snapshots can be used to estimate the ideal covariance matrix. Due to the stationarity of the signal, the sample covariance matrix will converge to the ideal covariance matrix with probability 1 when the number of snapshots tends to be infinite. That ***R_xx_*** is Hermitian matrix determines that its principal diagonal contains the power information instead of the angle information of signals. Therefore, the principal diagonal entries are replaced by 0, and the imaginary part of the upper triangular is taken, and the real part of the lower triangular is taken. After norm normalization processing, the *M* × *M* real matrix can be utilized as the input of the DOA estimation network. The power information of incoming signals is the key to monitor the effectiveness of incoming signals, and therefore the normalized *M* × *M* real matrix retaining the principal diagonal can be utilized as the input of the monitoring network.

### 3.2. Neural Network Architecture

#### 3.2.1. Signal Monitoring Network Architecture

The purple part of Figure 7 is the signal monitoring network, which consists of four layers. The first layer is the input one, and the input is an *M* × *M* real number matrix that can be from any ULA of the UTA. 

The second layer is an *M* × 1 max pooling layer [33] with the stride of 1. The third layer is a 1 × *M* average pooling layer [33] with the stride of 1. The fourth layer is the output one with only one neuron, and the activation function adopts sigmoid [20]. The cost function of the network adopts cross-entropy, which can be expressed as:(11)$$C(w,b)=-\frac{1}{m}\sum_{i=1}^{m}\left[y_{i}\log{\hat{y}}_{i}+(1-y_{i})\log(1-{\hat{y}}_{i}\right)],$$
where *m* denotes the total number of samples, and $y_{i}$ denotes the ground truth label, and ${\hat{y}}_{i}$ denotes the network output, and *w* denotes the network weight, and *b* denotes the network bias. 

#### 3.2.2. 1D-DOA Network Architecture

The orange, green, blue, and grey parts of Figure 7 are four 1D-DOA networks with identical parameters, which constitute the DOA estimation network in parallel. In the design phase, the 1D-DOA network architecture fails to be determined in one step. We have made the following attempts on the 1D-DOA network architecture: (1) the numbers of layers of the neural network are sequentially set from 8 to 18; (2) the numbers of filters in each CNN layer are set from 2 to 32; (3) the sizes of CNN filters are set to 2 × 2, 3 × 3, 4 × 4, and 5 × 5; (4) a max pooling layer or average pooling layer is introduced following each CNN layer; (5) Long Short-Term Memory (LSTM) layer is introduced before the full connection layer and the number of output units of the LSTM layer is changed; (6) the numbers of neurons in the fully connected layer are set to 2, 4, 8, 16, 32 and 64. On the premise of ensuring the estimation accuracy, we minimize the network parameters and finally determine the 1D-DOA network architecture.

The 1D-DOA network consists of 14 layers. The first layer is the input one, and the input is an *M* × *M* real matrix, which is generated by the UTA output. The second to eleventh layers are CNN ones, which are composed of 32, 32, 16, 16, 8, 8, 4, 4, 2, and 2 filters in turn. The size of the filters is set to 3 × 3. The convolution stride is all set to 1. The padding mode is all set to same. The activation functions adopt ReLU [34]. The twelfth and thirteenth layers, with 64 and 32 neurons, respectively, are fully connected ones, and the activation functions adopt ReLU. The fourteenth layer, with one neuron, is the output one, and the activation function adopts ReLU. The cost function of the network adopts the mean squared error (MSE), which can be expressed as:(12)$$C(w,b)=\frac{1}{m}\sum_{i=1}^{m}{\left(\theta_{i}-{\hat{\theta }}_{i}\right)}^{2},$$
where *m* denotes the total number of samples, and $\theta_{i}$ denotes the ground truth label, and ${\hat{\theta }}_{i}$ denotes the network output, and ***w*** denotes the network weight, and ***b*** denotes the network bias.

This study formulates DOA estimation as a regression problem. If DOA estimation is formulated as a classification problem, the outputs of neural networks will be discrete. Accurate results could be obtained in the case of high SNR and low angular resolution. However, when the SNR is low, or the angular resolution is high, various angles will overlap in a multi-dimensional space due to the presence of noises. The overlap of angles makes it difficult to get higher accuracy, and resulting in the inconvenience of evaluating the network architecture through the accuracy. In addition, the architecture of classification networks fundamentally limits the reduction of DOA estimation errors. Assume that there is a classification network with 181 neurons in the output layer to classify 0°–180° (1° resolution). When the observed angles are integers, the estimation accuracy of the classification network is 100%. Even so, the continuous real angles would bring about the possibility that the maximum angle error of the classification network reach 0.5°. The proposed 1D-DOA network maps the signal input features to angles and outputs continuous values, which could fundamentally reduce DOA estimation errors.

### 3.3. Neural Network Training

The neural network training process is conducted using simulated data created in Matlab. The sensor spacing of the UTA is set to 0.48 times as long as the signal wavelength. The incident angles of EM signals in training sets and validation sets lie between [0°, 180°] (1° resolution), and the total number of classes is 181. Inspired by Chakrabarty et al. [35], the signal amplitude of all data is randomly generated to enhance the robustness of the neural network to the signal amplitude. This study explores glorot uniform [36] method to initiate the weight matrix of the network, and the initial value of the bias matrix is 0. Furthermore, this study explores Adam [37] in the backpropagation, and the mini-batch size [38] is set to 1024.

#### 3.3.1. Signal Monitoring Network Training

The results of analysis depict that, on the one hand, when SNR = −15 dB, the MSE of the 1D-DOA network output is about 3 degree^2^, and we deem the error is acceptable; on the other hand, when SNR = −16 dB, the MSE is about 7 degree^2^, and the error is significant (See Figure 13a in Section 4.2.1 for details). Therefore, when SNR = −15 dB, 10 pieces of data are generated for each angle, and the data label is 1, which indicates that the signal is effective; when SNR = −16 dB, 10 pieces of data are generated for each angle, and the data label is 0, which indicates that the signal is invalid. A total of 3620 pieces of data constitute the training set. 

In order to fully validate the performance of the monitoring network, the SNR ranges from −25 dB to −16 dB (1 dB step), and 10 pieces of data, with the label of 0, are generated for each angle under each SNR; the SNR ranges from −15 dB to 20 dB (1 dB step), and 10 pieces of data, with the label of 1, are generated for each angle under each SNR. A total of 83,260 pieces of data constitute the validation set. The training set and validation set are generated independently to ensure no duplicate data.

Figure 8 exhibits the training process of the monitoring network. The learning rate is set to 0.5, and the training epoch is set to 500. The performance of the validation set is obviously better than that of the training set, which is rare in general neural network training and is caused by the composition of the data set. The training set consists of −15 dB and −16 dB data, which are the SNR thresholds of the signal monitoring. Each data in the training set may contribute to the increase of the cost function, and the accuracy is not very high. However, data of −15 dB and −16 dB account for merely 1/23 of the validation set, and the data of other SNR, which deviate from the SNR thresholds, should make a tiny contribution to the increase of the cost function and achieve high accuracy. Therefore, in Figure 8, the cost of the validation set is lower than that of the training set, and the accuracy of the validation set is higher than that of the training set. In addition, we should note that the appropriate training set can be designed for the monitoring network according to actual DOA estimation error requirements.

#### 3.3.2. 1D-DOA Network Training

Figure 9 displays the training process of the 1D-DOA network. When SNR = −15 dB and SNR = 10 dB respectively, 20 pieces of data with the labels of ground truth are generated to fit DOA estimation in the scenario of low SNR and prevent over-fitting for each angle because of disturbance of the ULA output covariance matrices from underlying noises. A total of 7240 pieces of data constitute the training set. In order to fully validate the performance of the 1D-DOA network, the SNR ranges from −15 dB to 20 dB (1dB step), and 10 pieces of data, with the labels of ground truth, are generated for each angle under each SNR. A total of 65,160 pieces of data constitute the validation set. The training set and validation set are generated independently to ensure no duplicate data. The learning rate is set to 0.001, 0.0005, 0.0001, 0.00005, and 0.00001 in turn, and the epoch for each learning rate is set to 600. The reason why the validation set cost in Figure 9 is lower than the training set cost is similar to that in Figure 8.

### 3.4. Post-Processing

From the DOA estimation network, we could obtain *θ_CA_*, *θ_CB_*, *θ_BA_*, and *θ_BC_*. As shown in Figure 10a, in the space, *θ_CA_* can determine the cone *C_CA_* in which *s* locates, and *θ_CB_* can determine the cone *C_CB_* in which *s* locates. *C_CA_* and *C_CB_* can determine the line *L_C_* in which *s* and *C* locate. Similarly, *θ_BA_* and *θ_BC_* can determine the line *L_B_* in which *s* and *B* locate. As shown in Figure 10b, *s*, the intersection point of the two lines, could be estimated from the four angles. 

In practical implementation, it is unnecessary for *L_C_* and *L_B_* to have the intersection point due to the 1D estimation errors, and therefore we estimate the target position by searching the closest point to the two lines.

According to the spatial position (*x*, *y*, *z*) of EM signals, we derive the expression of *L_C_* from *θ_CA_* and *θ_CB_*, as depicted in Table 2. Similarly, we could obtain the expression of *L_B_*.

Where $C_{zy}=\frac{3\sin^{2}\theta_{CB}}{{\left(2\cos\theta_{CA}-\cos\theta_{CB}\right)}^{2}}-1$, $C_{zx}=\frac{1}{\cos^{2}\theta_{CB}}-1-\frac{1}{3}{\left(\frac{2\cos\theta_{CA}}{\cos\theta_{CB}}-1\right)}^{2}$, $C_{yx}=\frac{2\sqrt{3}\cos\theta_{CA}}{3\cos\theta_{CB}}-\frac{\sqrt{3}}{3}$

### 3.5. Additional Preprocessing in Non-Gaussian Noise

In the presence of non-Gaussian noise, the additional preprocessing presented in this section precedes the preprocessing presented in Section 3.1. The strong impulse of non-Gaussian noise at some moments would prevent ***R_xx_*** from converging and the neural network from obtaining appropriate input features. However, after normalizing the array output data with the infinity-norm, the amplitude of array output is limited to [0, 1], which can ensure the convergence. In a snapshot, the infinity-norm of any ULA output is defined as:(13)$${\left\|x(n)\right\|}_{\infty }=\underset{1\leq i\leq M}{\max}\left|x_{i}(n)\right|.$$

After normalizing the array output with the infinity-norm, the observation vector can be expressed as:(14)$$x_{\infty }(n)=\frac{1}{{\left\|x(n)\right\|}_{\infty }}x(n).$$

The observation vector expressed by Equation (8) can be rewritten as:(15)$$x_{\infty }(n)=As_{\infty }(n)+e_{\infty }(n),$$
where $s_{\infty }(n)=s(n)/{\left\|x(n)\right\|}_{\infty }$, and $e_{\infty }(n)=e(n)/{\left\|x(n)\right\|}_{\infty }$. The observation matrix expressed by Equation (9) can be rewritten as:(16)$$x_{\infty }=As_{\infty }+e_{\infty },$$
where $x_{\infty }=x\Lambda_{\infty }$, $s_{\infty }=s\Lambda_{\infty }$, $e_{\infty }=e\Lambda_{\infty }$, and $\Lambda_{\infty }=\mathrm{diag}[1/{\left\|x(1)\right\|}_{\infty },1/{\left\|x(2)\right\|}_{\infty },\cdots ,1/{\left\|x(N)\right\|}_{\infty }]$. The sample covariance matrix expressed by Equation (10) can be rewritten as:(17)$$\begin{array}{cc} R_{xx\infty } & =E\{x_{\infty }(n)x_{\infty }^{H}(n)\}\\ & =E\{[As_{\infty }(n)+e_{\infty }(n)]{\left[As_{\infty }(n)+e_{\infty }(n)\right]}^{H}\}\\ & =E\{As_{\infty }(n)s_{\infty }^{H}(n)A^{H}+e_{\infty }(n)s_{\infty }^{H}(n)A^{H}+As_{\infty }(n)e_{\infty }^{H}(n)+e_{\infty }(n)e_{\infty }^{H}(n)\}\\ & =E\{As_{\infty }(n)s_{\infty }^{H}(n)A^{H}+e_{\infty }(n)e_{\infty }^{H}(n)\}\\ & =AP_{\infty }A^{H}+\sigma_{e\infty }^{2}I \end{array}$$
where $P_{\infty }=E\{s_{\infty }(n)s_{\infty }^{H}(n)\}$, and $\sigma_{e\infty }^{2}$ can be viewed as the power of non-Gaussian noise normalized by infinity-norm. Note that ***R_xx_***_∞_ has a similar structure to ***R_xx_***.

Normalizing with the infinity-norm is equivalent to suppressing impulsive outliers in the presence of non-Gaussian noise and can ensure the convergence of ***R_xx_***_∞_. In addition, it should be noted that the smaller the characteristic exponent, the stronger the impulse, and the more attenuation to ***s***. That is equivalent to the decrease of the signal power.

## 4. Simulation Results

In this section, in order to better evaluate the proposed method, we not only analyze the performance of the signal monitoring network, but also analyze the performance of the DOA estimation network from the perspective of 1D and 3D. The simulation conditions are as follows: (1) the sensor spacing of the UTA is set to 0.48 times as long as the signal wavelength; (2) the signal amplitude of all test data is randomly generated to test the robustness of the neural network to the signal amplitude; (3) the carrier frequency is set to 100 MHz. Each simulation experiment is performed in the presence of Gaussian and non-Gaussian noise, respectively. In addition, due to the infinite variance of *α*-stable distribution for 0 < *α* < 2, an effective alternative SNR is defined in the presence of non-Gaussian noise, namely, the generalized signal-to-noise ratio (GSNR), which is utilized to evaluate the rate of the signal power over noise dispersion by $\mathrm{GSNR}=10\mathrm{lg}(\sigma_{s}^{2}/\gamma )$ [14].

### 4.1. Signal Monitoring Network Performance

In order to evaluate the performance of the monitoring network, we define the signal detection rate (SDR) as *N_E_*/*N_T_*. *N_E_* denotes the number of effective signals judged by the network, and *N_T_* denotes the total number of signals.

#### 4.1.1. Performance in Gaussian Noise

Figure 11a reveals the relationship between the SDR and SNR of the monitoring network. The SNR of the test set ranges from −25 dB to 20 dB (1 dB step), and 500 pieces of data are randomly generated in the range of [0°, 180°] (0.01° resolution) under each SNR. In the process of designing the training set, the signal with the SNR of −15 dB was set to be effective, and the signal with the SNR of −16 dB was set to be invalid. Therefore, Figure 11a shows SDR’s noticeable jump accompanying the SNR of about −15 dB and validates the effectiveness of the monitoring network.

In order to intuitively portrait the performance of the monitoring network, 200 pieces of data are randomly generated with the SNR ranging from −25 dB to 20 dB (0.01 dB step) and the angle ranging from 0° to 180° (0.01° resolution). Figure 11b displays a clear boundary between red dots and black dots when the SNR is about −15 dB. The red dots indicate that the test result is 1, and the signal is effective. The black dots indicate that the test result is 0, and the signal is invalid.

#### 4.1.2. Performance in Non-Gaussian Noise

Furthermore, we perform similar experiments in the presence of non-Gaussian noise. The GSNR of the test set ranges from −25 dB to 20 dB (1 dB step), and 500 pieces of data are randomly generated in the range of [0°, 180°] (0.01° resolution) under each GSNR. Figure 12a reveals the relationship between the SDR and GSNR of the monitoring network under different characteristic exponent. 

The relationship suggests that as the characteristic exponent decreases, the GSNR corresponding to the jump increases, since the smaller the characteristic exponent, the more the number and the larger the amplitude of outliers. After the array output is processed with the infinity-norm, the signal attenuation becomes larger. The larger attenuation requires higher corresponding GSNR to ensure the effectiveness of the signal.

The characteristic exponent is set to 1.0, and 200 pieces of data are randomly generated with the GSNR ranging from −25 dB to 20 dB (0.01 dB step) and the angle ranging from 0° to 180° (0.01° resolution). When GSNR is about −5 dB, the boundary in Figure 12b is consistent with the jump in Figure 12a.

### 4.2. 1D-DOA Network Performance

The accuracy of 1D-DOA estimation is the key to the proposed 3D-DOA estimation model. Thus, we must evaluate the performance of the 1D-DOA network, which is evaluated by the MSE and processing time. The MSE has been defined in Section 3.2.2.

#### 4.2.1. Performance in Gaussian Noise

Some simulations are carried out to compare the performance of the 1D-DOA network with that of other methods, including MUSIC, MVDR, TLS-ESPRIT, and RBF. The search step of MUSIC and MVDR is set to 0.01°, and the point corresponding to the spectral peak is the estimated angle. The training set and validation set of the RBF network are the same as those of the 1D-DOA network. The spread of the training set is searched in the range of [0.1, 20], and the desired MSE is searched in the range of [0.1, 5]. Finally, the validation set performs best when the spread is set to 4.9, and the MSE is set to 3. At this point, the MSE of the validation set of the RBF network is about 4.5 degree^2^. Other simulation conditions of these five methods are the same.

The SNR of the test set ranges from −15 dB to 10 dB (1 dB step), and 500 pieces of data are randomly generated in the range of [0°, 180°] (0.01° resolution) under each SNR. Figure 13a reveals the relationship between the MSE and SNR of each method. The 1D-DOA network is even better than the MUSIC algorithm. Although the MUSIC algorithm breaks through the Rayleigh limit and is close to the Cramer Rao bound, the search step still limits its accuracy. Strictly speaking, the final result of the MUSIC algorithm is still discrete, and therefore in order to ensure the accuracy of the MUSIC algorithm, the angle sampling step and search step of the test set are both set to 0.01°. However, the output of the proposed 1D-DOA network is continuous, so the accuracy is slightly better than that of the MUSIC algorithm. Figure 13a also reveals that the 1D-DOA network is superior to the RBF network. Although RBF can accurately fit functions, its generalization ability is not as good as that of CNN in terms of DOA estimation. In addition, since the angles are randomly sampled for each SNR, the MSE of each method in Figure 13a does not decrease monotonically with the increase of the SNR.

In order to intuitively portrait the performance of the 1D-DOA network, the SNR of the test set ranges from −15 dB to 10 dB (5 dB step), and 20 pieces of data are randomly generated in the range of [0°, 180°] (0.01° resolution) under each SNR. Figure 13b displays the correlation between observed angles and estimated angles. The Pearson product-moment correlation coefficient (*r_ppm_*) is 0.9999.

To highlight the superiority of the 1D-DOA network, we now compare this method with MUSIC, MVDR, TLS-ESPRIT, and RBF in terms of the processing time. The processing time of the 1D-DOA network and RBF network includes network running and data preprocessing time. The processing time of the 1D-DOA network is recorded as a unit time. Table 3 shows the results from 500 Monte Carlo run. The MUSIC algorithm and MVDR algorithm need longer processing time for the spectral peak search, while the TLS-ESPRIT algorithm and RBF need shorter processing time because of the avoidance of the spectral peak search. The estimation accuracy of the 1D-DOA network is much higher than that of the TLS-ESPRIT algorithm and RBF network, although the 1D-DOA network does not have distinct advantages in terms of processing time.

#### 4.2.2. Performance in Non-Gaussian Noise

The performance of the 1D-DOA network is also tested in the presence of non-Gaussian noise. The GSNR of the test set ranges from −5 dB to 20 dB (1 dB step), and 500 pieces of data are randomly generated in the range of [0°, 180°] (0.01° resolution) under each GSNR. Figure 14a reveals the relationship between the MSE and GSNR of the 1D-DOA network under different characteristic exponent. The relationship suggests that the performance is getting better and better with the increase of the GSNR or characteristic exponent. Still, DOA estimation is a difficult problem for the scenario of *α* = 0.1.

FLOM and PFOM are classic DOA estimation algorithms in the presence of non-Gaussian noise, but reference [13,14] has proved that CRCO and COBU are superior to FLOM and PFOM. In order to measure the performance of the 1D-DOA network, we utilize the corentropy-based correlation of CRCO and the corentropy-based operator of COBU to make a comparison with the 1D-DOA network, respectively. The search step of CRCO and COBU is set to 0.01°, and the point corresponding to the spectral peak is the estimated angle. The scale factor of CRCO is set to 1.4, and the parameter *μ* is set to 0.5. The weight factor and kernel size of COBU are set to 1. Other simulation conditions are the same as that of the 1D-DOA network. The GSNR ranges from −5 dB to 20 dB (1 dB step) with the characteristic exponent of 1.3, and 500 pieces of data are randomly generated in the range of [0°, 180°] (0.01° resolution) under each GSNR. Figure 14b reveals the relationship between the MSE and GSNR of each method, and the performance of the 1D-DOA network is significantly better than that of CRCO and COBU.

Then, the GSNR ranges from −5 dB to 20 dB (5 dB step) with the characteristic exponent of 1.3, and 20 pieces of data are randomly generated in the range of [0°, 180°] (0.01° resolution) under each GSNR. After processing by the 1D-DOA network, Figure 14c displays the correlation between observed angles and estimated angles. The Pearson product-moment correlation coefficient (r_ppm_) is 0.9999.

We also compare the processing time of the 1D-DOA network, CRCO, and COBU. The processing time of the 1D-DOA network includes network running and data preprocessing time, and it is recorded as a unit time. Table 4 shows the results from 500 Monte Carlo run. Because CRCO and COBU also need the spectral peak search, their processing time is longer than that of the 1D-DOA network.

### 4.3. 3D-DOA Estimation Performance

In order to evaluate the performance of the 3D-DOA estimation, we define the MSE_3D_ as:(18)$${\mathrm{MSE}}_{3D}=\frac{1}{3m}\sum_{i=1}^{m}\left[{\left(x-{\hat{x}}_{i}\right)}^{2}+{\left(y-{\hat{y}}_{i}\right)}^{2}+{\left(z-{\hat{z}}_{i}\right)}^{2}\right],\;$$
where *m* denotes the total number of samples, and (*x*, *y*, *z*) denotes the observed coordinate, and (${\hat{x}}_{i}$, ${\hat{y}}_{i}$, ${\hat{z}}_{i}$) denotes the estimated coordinate.

#### 4.3.1. Performance in Gaussian Noise

In this section, we compare the proposed method based on the UTA with the 3D-MUSIC algorithm based on the uniform circular array (UCA) [31,32]. In order to facilitate sampling and ensure the accuracy of the 3D-MUSIC algorithm, the position of EM signals is limited in the space 200 to 300 [m] away from the origin of Cartesian coordinate system, and the coordinate *z* is more than 30 [m]. Random sampling is carried out in the spherical coordinate system. Considering that the 3D-MUSIC algorithm needs spectral peak search, too small search step will lead to an enormous amount of computation. In this case, the orders of magnitude of the angle sampling and angle search step are set to 1°, and the orders of magnitude of the radial distance sampling and radial distance search step are set to 1 [m]. The proposed method avoids the problem of massive computation, and the actual angles and radial distances are continuous. In this case, the order of magnitude of the angle sampling is set to 0.01°, and the order of magnitude of the radial distance sampling is 0.01 [m]. The number of sensors of the UCA is the same as that of the UTA, and the array radius is 4.5 times as long as the signal wavelength. Other simulation conditions are the same as that of the proposed method.

The SNR ranges from −5 dB to 20 dB (1 dB step), and 500 positions are randomly sampled for each SNR in the space. Figure 15a reveals the relationship between the MSE_3D_ and SNR of the proposed method and 3D-MUSIC algorithm. Furthermore, the radial distance ranges from 200 to 300 [m] (10 [m] step) with the SNR of 15 dB, and 500 positions are randomly sampled for each radial distance in the space. Figure 15b compares the MSE_3D_ of the proposed method and 3D-MUSIC algorithm versus the radial distance. The proposed method outperforms the 3D-MUSIC algorithm under stricter sampling conditions. In addition, the real angles and radial distances are continuous, but the angle search step of the 3D-MUSIC algorithm is set to 1°, and the radial distance search step is set to 1 [m]. The search step will be greatly limited in practical applications. Even so, the processing time of the 3D-MUSIC algorithm is still about 70,000 times as long as that of the proposed method, and such a large amount of computation is difficult to process in real time.

In order to intuitively portrait the 3D-DOA estimation performance of the proposed method, the SNR is set to 20 dB, and 100 points are randomly sampled in the space. Figure 15c suggests that observed positions and estimated positions basically coincide.

#### 4.3.2. Performance in Non-Gaussian Noise

The space range setting has been described in Section 4.3.1. The GSNR ranges from 0 dB to 25 dB (1 dB step), and 500 positions are randomly sampled for each GSNR in the space. Figure 16a reveals the relationship between the MSE_3D_ and GSNR of the proposed method under different characteristic exponent. Due to the effect of the 1D-DOA estimation accuracy, the performance of the proposed method is getting better and better with the increase of the GSNR or characteristic exponent.

To the best of our knowledge, there are no published 3D-DOA estimation algorithms in the presence of non-Gaussian noise. In order to validate that the proposed method is superior to conventional algorithms, we have modified CRCO and COBU based on the UCA. The array output covariance matrix of the 3D-MUSIC algorithm is replaced by the correentry-based correlation and correntropy-based operator, respectively. The 3D-DOA estimation using CRCO or COBU is achieved and compared with the proposed method. Considering the computational complexity, the data sampling and search step of the modified CRCO and COBU are the same as that of the 3D-MUSIC algorithm. The UCA setting has been described in Section 4.3.1. Other simulation conditions are the same as that of the proposed method. The GSNR ranges from 0 dB to 25 dB (1 dB step) with the characteristic exponent of 1.6, and 500 positions are randomly sampled for each GSNR in the space. Figure 16b reveals the relationship between the MSE_3D_ and GSNR of each method. Furthermore, the radial distance ranges from 200 to 300 [m] (10 [m] step) with the GSNR of 20 dB and the characteristic exponent of 1.6, and 500 positions are randomly sampled for each radial distance in the space. Figure 16c compares the MSE3D of each method versus the radial distance. Obviously, the proposed method outperforms the modified CRCO and COBU. In addition, the processing time of the modified CRCO and COBU is about 2,350 times as long as that of the proposed method.

In order to intuitively show the 3D-DOA estimation performance of the proposed method in the presence of non-Gaussian noise, the GSNR is set to 25 dB, and the characteristic exponent is set to 1.6, and 100 points are randomly sampled in the space. Figure 16d displays the 3D-DOA estimation result.

In addition, in order to study the generalization ability of the infinity-norm normalization to DOA estimation in Gaussian noise, employing the infinity-norm normalization, we preprocess the UTA output data, which generates Figure 11, Figure 13 and Figure 15, and then regenerate these figures. The shapes of these figures are basically unchanged. Moreover, in the presence of impulse noise, usually Gaussian noise is also present. After the verification of simulation experiments, infinity-norm normalization preprocessing can still be generalized to DOA estimation in this case.

## 5. Conclusions

In this paper, we presented a novel 3D-DOA estimation method of stationary EM signals using CNN in Gaussian or non-Gaussian noise environment. To do this, we proposed the 1D-DOA network using CNN in the presence of Gaussian noise. We have shown that the proposed method overcame the high computational cost of conventional DOA estimation methods. The signal monitoring network was also proposed to address the possible misjudgment problem of the network. The UTA was utilized to achieve 3D-DOA estimation of EM signals. In the presence of non-Gaussian noise, the infinity-norm was utilized to normalize the UTA output. We have shown that the infinity-norm normalization effectively suppressed the impulsive outliers, and the 3D-DOA estimation of EM signals was also achieved. Finally, we proved its superiority and effectiveness by comparing our proposed method with the existing methods.

The experiments reached the following conclusions: (1) the signal monitoring network can effectively control the output of the 3D-DOA estimation model; (2) in the presence of Gaussian noise, the proposed method is superior to several existing methods in the computation speed and accuracy of 1D-DOA and 3D-DOA estimations; (3) in the presence of non-Gaussian noise, the infinity-norm normalization preprocessing can provide appropriate input features for neural networks, and neural networks can perform DOA estimation well. In conclusion, CNN has better generalization ability in DOA estimation.

We should solve the following problems that still remain: (1) To improve the accuracy of 3D-DOA estimation by modifying post-processing mode; (2) To carry out study on non-stationary signals; (3) To implement the proposed approach in the real environment.

## Figures and Tables

**Figure 1 sensors-20-02761-f001:**
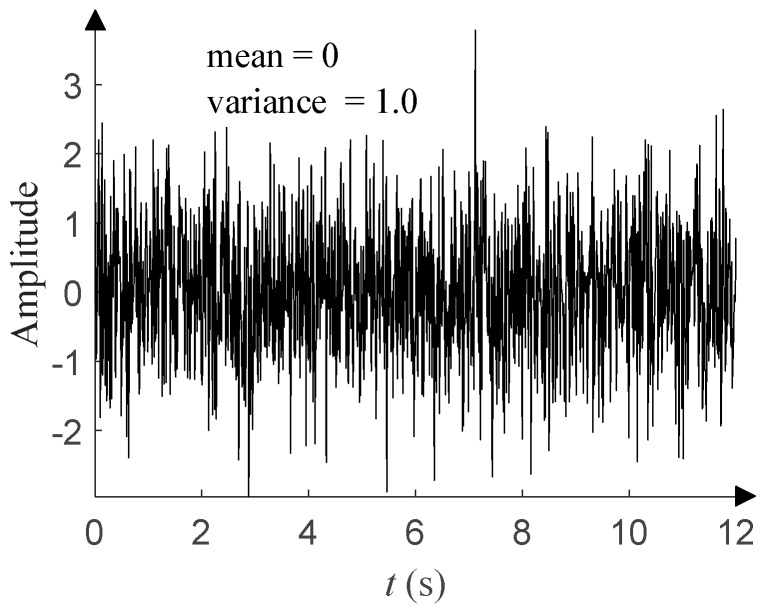
Time-domain waveform of Gaussian distribution.

**Figure 2 sensors-20-02761-f002:**
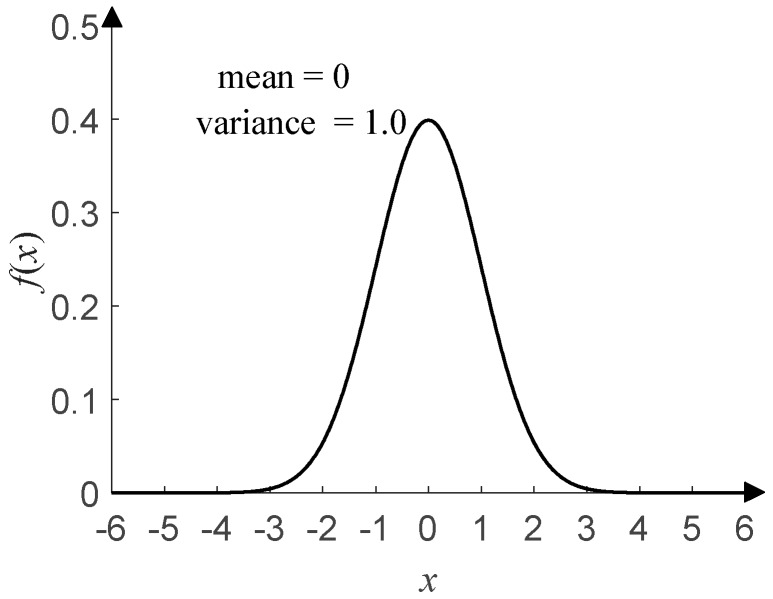
PDF of Gaussian distribution.

**Figure 3 sensors-20-02761-f003:**
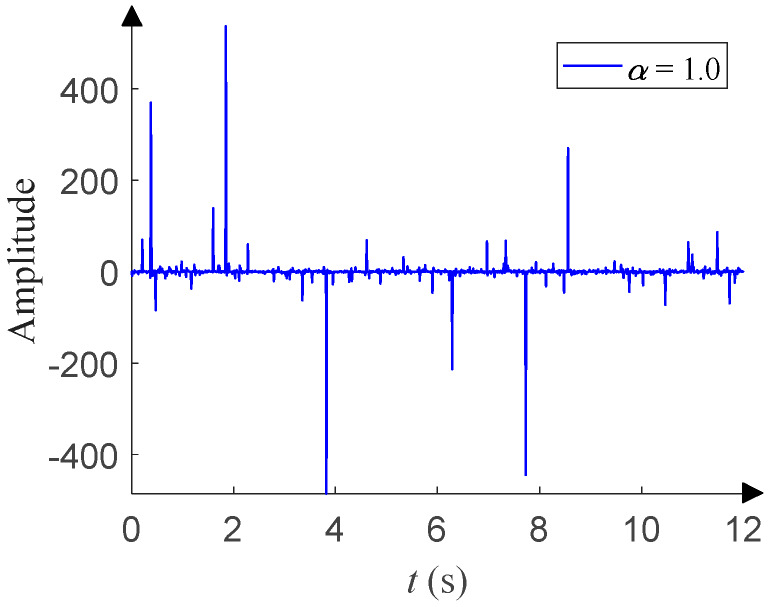
Time-domain waveform of *α*-stable distribution.

**Figure 4 sensors-20-02761-f004:**
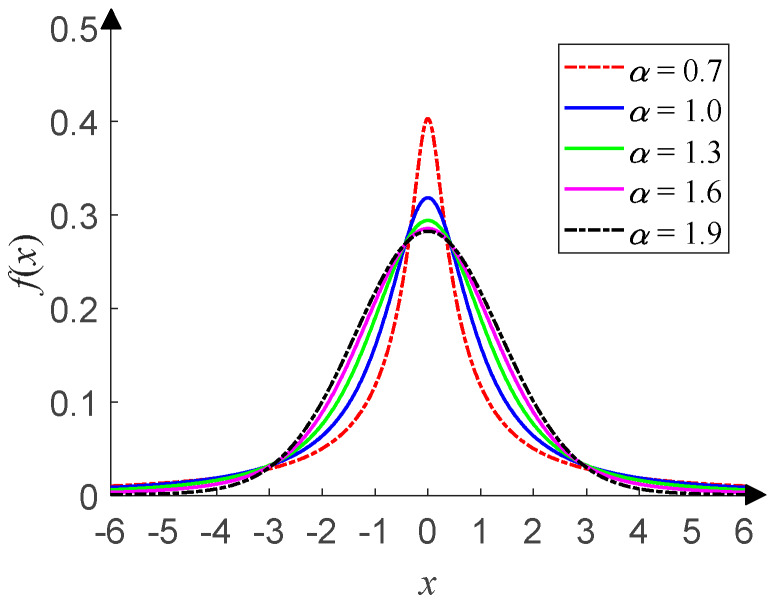
PDF of *α*-stable distribution.

**Figure 5 sensors-20-02761-f005:**
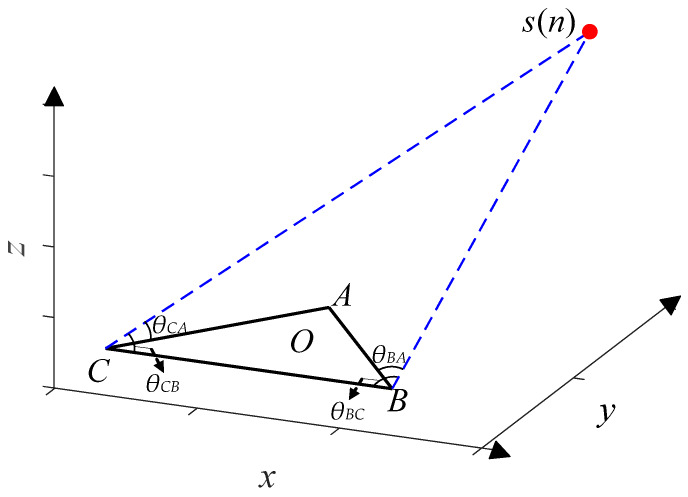
UTA model.

**Figure 6 sensors-20-02761-f006:**
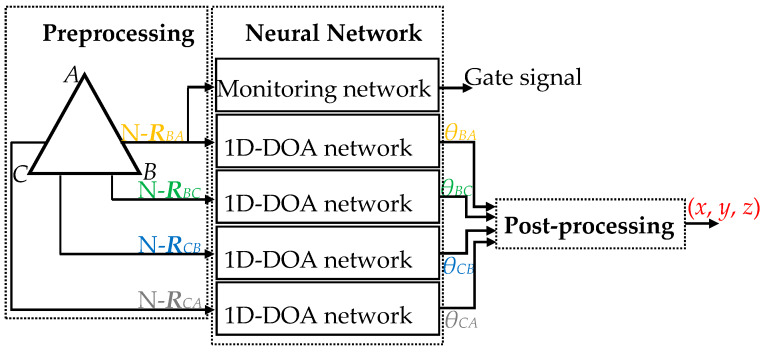
Block diagram of the 3D-DOA estimation model.

**Figure 7 sensors-20-02761-f007:**
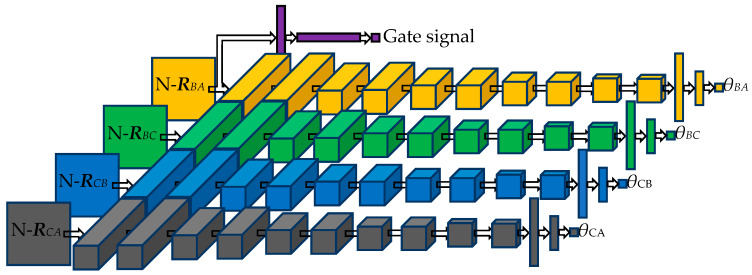
DOA estimation neural network architecture.

**Figure 8 sensors-20-02761-f008:**
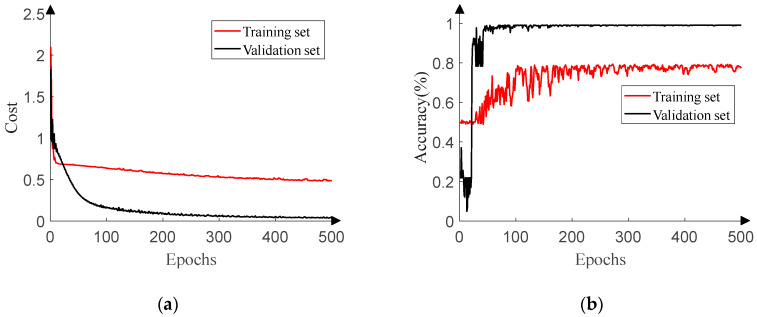
Signal monitoring network training process. (**a**) Training and validation cost. (**b**) Training and validation accuracy.

**Figure 9 sensors-20-02761-f009:**
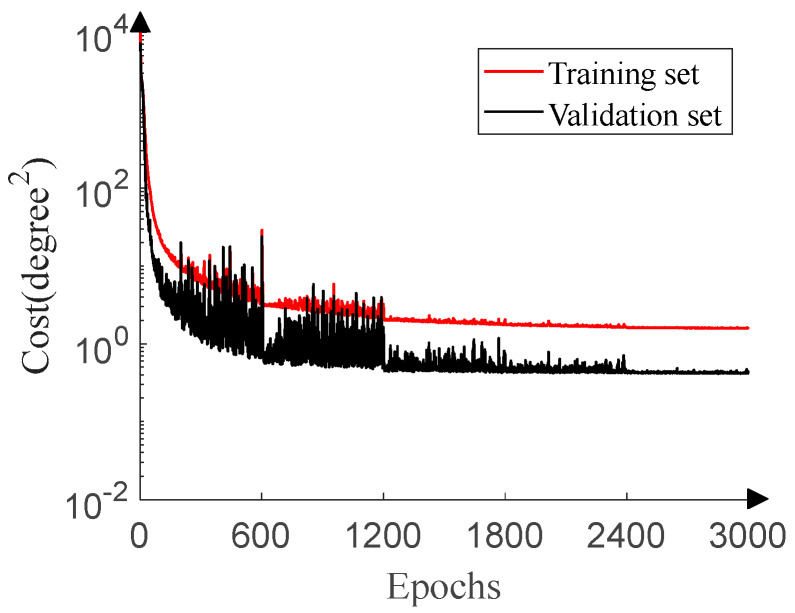
1D-DOA network training process.

**Figure 10 sensors-20-02761-f010:**
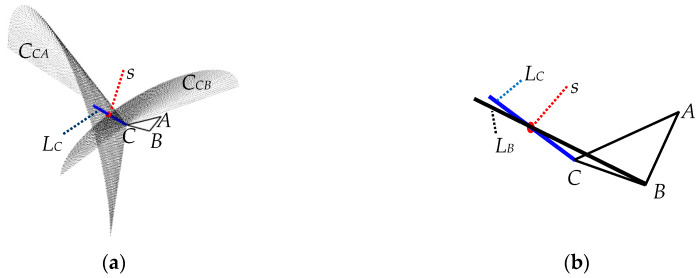
Schematic diagram of 3D-DOA estimation based on the UTA. (**a**) *L_C_* determined by *C_CA_* and *C_CB_*. (**b**) *s* determined by *L_C_* and *L**_B_*.

**Figure 11 sensors-20-02761-f011:**
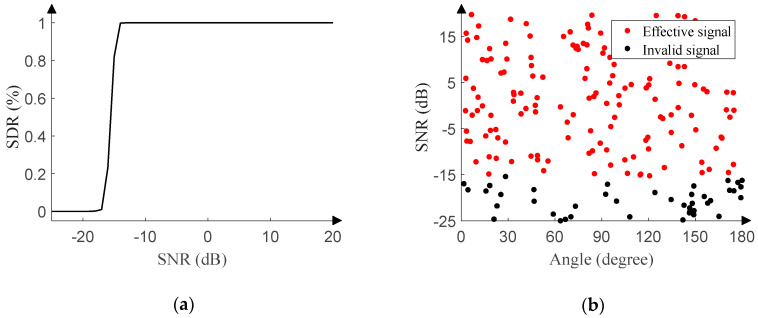
Signal monitoring network performance in Gaussian noise. (**a**) SDR versus SNR. (**b**) Response of the signal monitoring network.

**Figure 12 sensors-20-02761-f012:**
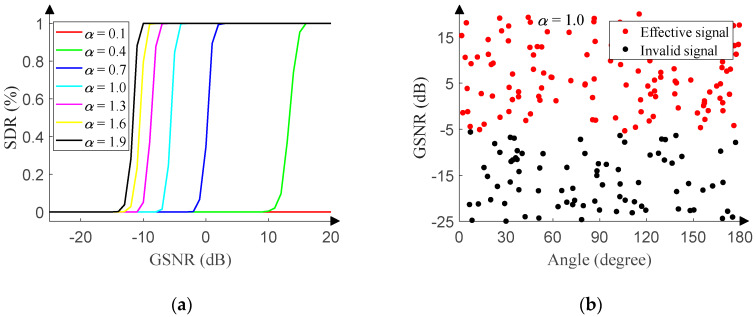
Signal monitoring network performance in non-Gaussian noise. (**a**) SDR versus GSNR. (**b**) Response of the signal monitoring network.

**Figure 13 sensors-20-02761-f013:**
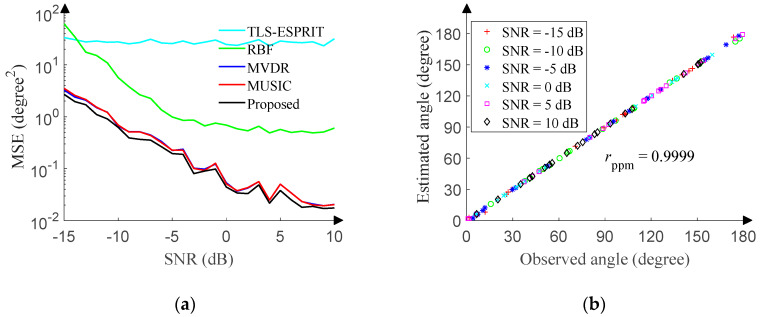
1D-DOA network performance in Gaussian noise. (**a**) MSE versus SNR. (**b**) Angle correlation diagram.

**Figure 14 sensors-20-02761-f014:**
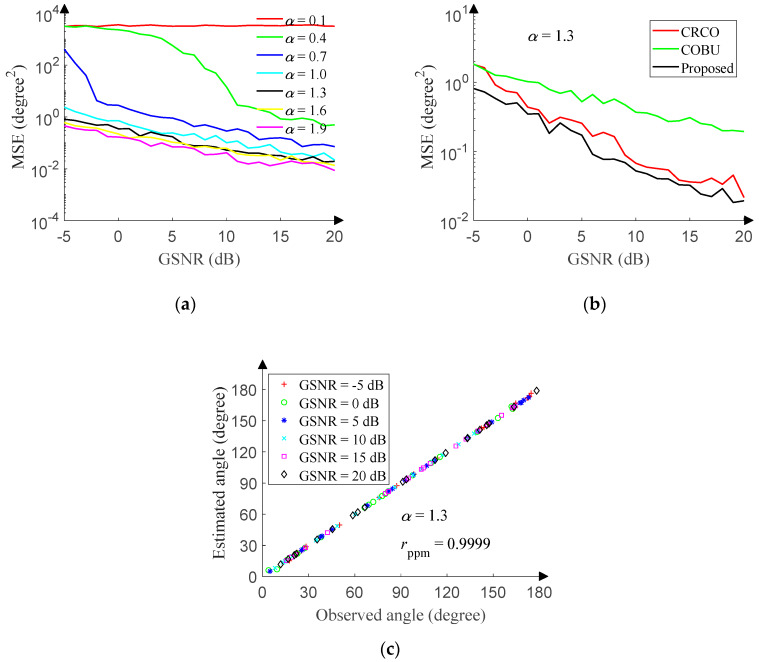
1D-DOA network performance in non-Gaussian noise. (**a**) MSE versus GSNR. (**b**) MSE versus GSNR. (**c**) Angle correlation diagram

**Figure 15 sensors-20-02761-f015:**
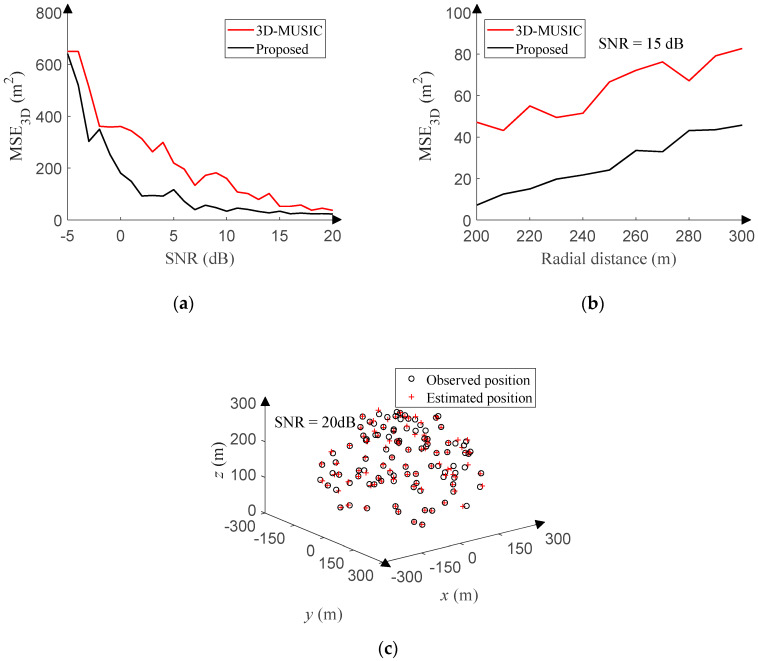
3D-DOA estimation performance in Gaussian noise. (**a**) MSE versus GSNR. (**b**) MSE versus radial distance. (**c**) Response of the proposed method.

**Figure 16 sensors-20-02761-f016:**
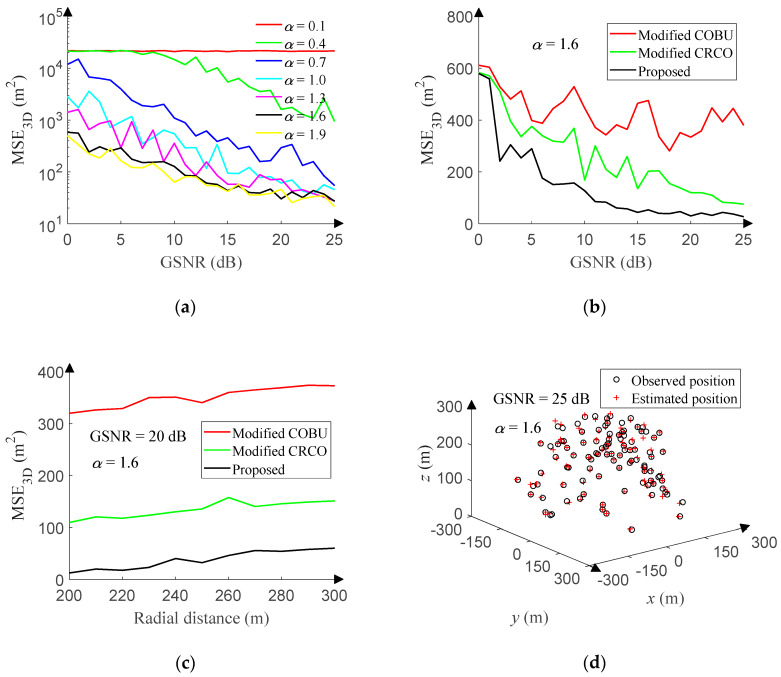
3D-DOA estimation performance in non-Gaussian noise. (**a**) MSE versus GSNR. (**b**) MSE versus GSNR. (**c**) MSE versus radial distance. (**d**) Response of the proposed method.

**Table 1 sensors-20-02761-t001:** Mathematical notations.

Notations	Explanations
(·)^T^	Transpose
(·)^H^	Conjugate transpose
∀	Arbitrary value
*E*{·}	Expectation operator
***I***	Identity matrix
‖·‖_∞_	Infinity-norm of a vector

**Table 2 sensors-20-02761-t002:** Expressions of *L*_C__._

$C_{zy}$	$\theta_{CA}$	$\theta_{CB}$	$C_{yx}$	*L_C_*
$C_{zy}>0$	$0^{{}^{\circ}}<\theta_{CA}<90^{{}^{\circ}}$	$0^{{}^{\circ}}<\theta_{CB}<90^{{}^{\circ}}$	$C_{yx}>0$	$z=\sqrt{C_{zy}}\left(y-C_{y}\right)=\sqrt{C_{zx}}\left(x-C_{x}\right)$
			$C_{yx}=0$	$z=\left(x-C_{x}\right)\tan\theta_{CB}$, $y=C_{y}$
			$C_{yx}<0$	$z=\sqrt{C_{zy}}\left(C_{y}-y\right)=\sqrt{C_{zx}}\left(x-C_{x}\right)$
	$90^{{}^{\circ}}<\theta_{CA}<180^{{}^{\circ}}$	$0^{{}^{\circ}}<\theta_{CB}<90^{{}^{\circ}}$	$C_{yx}=\forall$	$z=\sqrt{C_{zy}}\left(C_{y}-y\right)=\sqrt{C_{zx}}\left(x-C_{x}\right)$
	$90^{{}^{\circ}}<\theta_{CA}<180^{{}^{\circ}}$	$90^{{}^{\circ}}<\theta_{CB}<180^{{}^{\circ}}$	$C_{yx}>0$	$z=\sqrt{C_{zy}}\left(C_{y}-y\right)=\sqrt{C_{zx}}\left(C_{x}-x\right)$
			$C_{yx}=0$	$z=\left(x-C_{x}\right)\tan\theta_{CB}$, $\;y=C_{y}$
			$C_{yx}<0$	$z=\sqrt{C_{zy}}\left(y-C_{y}\right)=\sqrt{C_{zx}}\left(C_{x}-x\right)$
	$0^{{}^{\circ}}<\theta_{CA}<90^{{}^{\circ}}$	$90^{{}^{\circ}}<\theta_{CB}<180^{{}^{\circ}}$	$C_{yx}=\forall$	$z=\sqrt{C_{zy}}\left(y-C_{y}\right)=\sqrt{C_{zx}}\left(C_{x}-x\right)$
	$\theta_{CA}=90^{{}^{\circ}}$	$0^{{}^{\circ}}<\theta_{CB}<90^{{}^{\circ}}$	$C_{yx}=\forall$	$z=\sqrt{\frac{3}{\cos^{2}\theta_{CB}}-4}\left(C_{y}-y\right)=\sqrt{\frac{1}{\cos^{2}\theta_{CB}}-\frac{4}{3}}\left(x-C_{x}\right)$
		$90^{{}^{\circ}}<\theta_{CB}<180^{{}^{\circ}}$	$C_{yx}=\forall$	$z=\sqrt{\frac{3}{\cos^{2}\theta_{CB}}-4}\left(y-C_{y}\right)=\sqrt{\frac{1}{\cos^{2}\theta_{CB}}-\frac{4}{3}}\left(C_{x}-x\right)$
		$\theta_{CB}=90^{{}^{\circ}}$	$C_{yx}=\forall$	$x=C_{x}$, $y=C_{y}$, z=∀
	$0^{{}^{\circ}}<\theta_{CA}<90^{{}^{\circ}}$	$\theta_{CB}=90^{{}^{\circ}}$	$C_{yx}=\forall$	$z=\sqrt{\frac{3}{4\cos^{2}\theta_{CA}}-1}\left(y-C_{y}\right)$, $x=C_{x}$
	$90^{{}^{\circ}}<\theta_{CA}<180^{{}^{\circ}}$	$\theta_{CB}=90^{{}^{\circ}}$	$C_{yx}=\forall$	$z=\sqrt{\frac{3}{4\cos^{2}\theta_{CA}}-1}\left(C_{y}-y\right)$, $x=C_{x}$
$C_{zy}\leq 0$	$\theta_{CA}=\forall$	$\theta_{CB}=\forall$	$C_{yx}=\forall$	$y-C_{y}=\left(x-C_{x}\right)\left(\frac{2\sqrt{3}\cos\theta_{CA}}{3\cos\theta_{CB}}-\frac{\sqrt{3}}{3}\right)$, z=0

**Table 3 sensors-20-02761-t003:** Comparison of methods in processing time.

Methods	Processing Time
MUSIC [4,5]	152.89
MVDR [6,7]	100.92
TLS-ESPRIT [8,9,10]	1.82
RBF [29]	1.44
1D-DOA network	1

**Table 4 sensors-20-02761-t004:** Comparison of methods in processing time.

Methods	Processing Time
CRCO [13]	5.18
COBU [14]	4.80
1D-DOA network	1

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
