# Peer review of "A Novel Approach to 3D-DOA Estimation of Stationary EM Signals Using Convolutional Neural Networks"

_sensors, 2020, doi:10.3390/s20102761_

Round 1

Reviewer 1 Report

The authors present a Neural Network Approach for the classification of 3D-DOA estimation and Signal monitoring for Electromagnetic signals. It is important to notice that the estimation is with cartisian coordinates.  Their approach is based on Convolutional Networks. The authors present their model and a sequence of experiments, first within a training and validation second, and in a testing setting.

The formulation of the problem I consider very limiting since it is set as a stationary and it is not very interesting. I know it is the first step towards other settings, but I think as a field we should focus on non-stationary which is more challenging, if you think this is too out of your scope, I suggest the tittle reflects that the paper only address the stationary setting.

Overall I found the paper very informative. The following are the points I consider the authors should address:

  1. I think more should be said about the data used for training, validation and testing. In particular for training, it would be nice to provide their distribution. It should be clear how was obtained, and which cuadrants
  2. I wonder about the effects of distance of the source to the target, why this was not explored, the sensibility of the approach to the distance.
  3. I do not understood if the data was collected with real sensors or it was simulated. If the latest, I suggest to change the title to incorporate the notion that so far there this study is only simulations and does not use real data or scenarios.
  4. The second part of the experimentation makes it clear it is simulation. I will thank if more information about how the simulation was carry out.
  5. In case of the simulation, the noise was added to what kind of signal, was this static? dynamic?
  6. I'm confused why θBA is being calculated, if it seems to be easily defined by θBA - 60
  7. The networks are used a classifiers, this is very strong setting for the angular data, because it assummes independence among the angle possition and we know this is no the case. Why not use a regression layer instead?
  8. Figures 8 and 9 show something rarely seen, which are much better validation results than training. Generally shows that there is a validation data are depended of training, could be this possible? do you have an explanation why validation are too good compare to training?
  9. Four NN were used, why not use part of the same network? let the NN to learn to process the information in a similar way (lower CNN layers) and let the last layers to deal with the different outputs.

Finally, the major issue I have after reading the paper is to clarify if the data was simulated and how was obtained, because if it is the case; it could be that the network is learning the underlying model and I will no see major contribution, even adding noise it is not enough because the models can learn to extract this. It will only show that the NN might be able to do with real data but not if would do it at a good job since real data is much more noise and unpredictable. NN should be used when we do not know the underlying method, this is the case with collected real data.

Reviewer 2 Report

In this paper the authors introduced a new 3D-DOA method for estimating EM signals using CNN in Gaussian or non-Gaussian noise environment. To this end, the authors designed the 1D-DOA network using CNN in the presence of Gaussian noise. They then showed that the proposed method exceeded the high computational costs of conventional DOA estimation methods. UTA was used to achieve 3D-DOA estimation of EM signals.
The authors proposed a new method of a new 3D-DOA method for estimating EMG signal propagation. They described the method and proven it with the methods used so far. They designed a model of signal propagation estimation and neural network architecture that they underwent learning / training. Comparison tests were performed on the level of mathematical comparisons of several known methods.
The work is conceptually and text on a good professional level. From the formal comments I would recommend to the graphs Fig. 1-4, FIG. 8-9, Fig.11-16 to the x, y axes to add arrows of increasing magnitude.

I recommend the text for publication

Reviewer 3 Report

In general, this is a good written article. I have only minor remarks.

Abstract, lines 20-21. “… the proposed method outperforms several existing methods…”.

It is worth explicitly mentioning the methods with which the comparison was made.

In the introduction section, I think it’s worth adding the following publication.

Chakrabarty and E. A. P. Habets, "Broadband doa estimation using convolutional neural networks trained with noise signals," 2017 IEEE Workshop on Applications of Signal Processing to Audio and Acoustics (WASPAA), New Paltz, NY, 2017, pp.

136-140.

Page 2, line 98. It would be nice to give a formula for the alpha-stable distribution.
